# Dental wear and oral pathology among sex determined Early Bronze-Age children from Franzhausen I, Lower Austria

**Marlon Bas[1,2], Christoph Kurzmann[3], John Willman[4], Doris Pany-Kucera[5], Katharina Rebay-Salisbury●[1], Fabian Kanz●[2]***

1 Austrian Archaeological Institute, Austrian Academy of Sciences, Vienna, Austria, 2 Center for Forensic Medicine, Medical University of Vienna, Vienna, Austria, 3 Clinical Division of Conservative Dentistry and Periodontology, University Clinic of Dentistry, Medical University of Vienna, Vienna, Austria, 4 Laboratory of Prehistory, CIAS–Research Centre for Anthropology and Health, Department of Life Sciences, University of Coimbra, Coimbra, Portugal, 5 Department of Anthropology, Natural History Museum, Vienna, Austria

* fabian.kanz@meduniwien.ac.at

**Data Availability Statement:** All relevant data are within the paper.

**Funding:** This study was undertaken within the framework of the ERC project 'The value of

## Abstract

The physical properties of diet and oral health throughout childhood play an important role in the development of human dentition, and differed greatly before the industrial revolution. In this study we examined dental wear and oral pathology in a sample of children from the Early Bronze-Age to investigate the physical and mechanical properties of childhood diet and related oral health. We explore cross-sectional age and sex-based variation of children in the sample. The analysis was carried out on the dentitions of 75 children, 978 teeth, excavated from the Early Bronze-Age cemetery Franzhausen I in Lower Austria. Presence of dental caries and calculus was recorded. Dental wear was measured using dentine exposure, occlusal topography, and dental microwear texture analysis. Sex determination was carried out using amelogenin peptide analysis. Caries were found in only 4 individuals (crude prevalence rate—5%, 95% CI 1% to 13%), affecting only 5 teeth (true prevalence rate—less than 1%). Dentine exposure was observed in over 70% of deciduous molars and dental wear measurements indicate a comparatively strong dental wear accumulation especially, among younger children, when compared to modern-day and later pre-industrial populations. Microwear textures presented a high complexity (Asfc > 2)/low anisotropy (epLsar < 1) profile, especially in older children. Differences between male and female children were not generally significant but increased dentine exposure was observed in the lower molars of younger female children. Our results suggest that the Early Bronze-Age children at Franzhausen I consumed a non-cariogenic diet, more abrasive and inclusive of harder/polyhedral foodstuffs than present-day children and some later Medieval children. Differences in dental wear accumulation were observed between children within the population, but with minimal variation between the sexes mostly occurring among younger children.

mothers to society: responses to motherhood and child rearing practices in prehistoric Europe'. This project has received funding from the European Research Council (ERC) under the European Union's Horizon 2020 research and innovation program (grant agreement No 676828, PI Katharina Rebay-Salisbury). The funders had no role in study design, data collection and analysis, decision to publish, or preparation of the manuscript.

**Competing interests:** The authors have declared that no competing interests exist.

## Introduction

The amount and nature of dental wear and the prevalence of pathological conditions among the dental remains of children from before the Industrial Revolution differ from children in industrialized societies today. Indeed, diet, environment, and food preparation influence macroscopic and microscopic dental wear as well as the prevalence of several pathological conditions of the oral cavity among human populations [1–5]. In many regions of the world, the accumulation of abrasive dental wear among adults has decreased and the prevalence of tooth decay and erosive dental wear has increased since the Palaeolithic. These observations are explained by technology-related increases in extra-oral food processing that have resulted in less chewing, the introduction of novel environmental abrasives into food, and a simultaneous increase in sugar consumption [6–8]. The recent decrease in dietary hardness and masticatory load during childhood in conjunction with the increased mechanisation of food processing has been suggested as a factor behind the high malocclusion rates in present-day populations [9–12]. However, the diversity of pre-industrial childhood masticatory behaviour and load, reflected in dental wear, remains poorly documented. The analysis of dental wear and pathological conditions provides information about physical properties of childhood diet in past populations [13–15], as such, childhood dental wear has been documented in several past and present non-industrialized populations [16–19], with more recent studies also focused on reconstructing childhood dietary transitions and documenting intra-population childhood dietary variation using dental microwear analysis [14, 15, 20–22]. The description of dental macrowear, microwear and pathology among children in past populations is not just crucial to improve our understanding of diet and quality of life during childhood within the specific contexts studied, but also to understand past diversity of human dental wear and pathological conditions in comparison to present-day populations.

The first objective of this study is to document childhood macroscopic and microscopic dental wear as well as dental caries and dental calculus in non-adults buried at Franzhausen I—(a large cemetery excavated in Lower Autria in the early 80s), focused on individuals with preserved dentitions having died before the age of 12 years—to reconstruct and document aspects of childhood diet and oral health in an Early Bronze Age agriculturalist community from Central Europe.

Several studies of intra-population childhood diet based on isotope ratio analyses have also highlighted gendered differences in childhood diet in past populations [23, 24]. Strongly gendered burial practices, even among younger children in the context of Franzhausen I, suggest gender was an important component of childhood identity in the community [25]. Recent developments in amelogenin peptide-based sex determination provide a reliable means to determine the sex of non-adult human remains [26–28]. Such methods allow the further comparison of dental wear and the prevalence of dental pathological conditions among male and female children to discuss potential gendered differences in childhood diet among prehistoric populations.

The second objective is therefore to compare the described aspects of dental wear and pathology between male and female children within the population and discuss the possibility of gender-based differences in childhood dietary provisioning.

## Materials and methods

### Sample

The 75 individuals under the age of 12 in this study derive from the Early Bronze Age cemetery of Franzhausen I, Austria. The cemetery was excavated between 1981 and 1983 in the course

of large-scale road developments and comprises 716 published grave contexts [29]. The remains are stored at the Natural History Museum of Vienna, Austria, where they can be examined upon request. The individuals buried at Franzhausen I are understood to be members of a small, rural Bronze Age community that settled in the Traisen river valley close to the Danube, approximately 50 km west of Vienna. The cemetery was in use for 300 to 400 years and was radiocarbon dated from 2050 to 1680 cal. BC. Individuals were placed in the grave according to gender, with girls and women placed on the right side of the body, head south, whereas boys and men were placed on the left side of the body, head north. That gendered treatment of bodies extended to children has recently been confirmed via peptide-based sex determination [25]. Grave good selection is also gender based, and differences in the amount and value of grave goods suggest socio-economic inequalities within the community [30].

Age and sex estimations of the skeletal remains were performed [31, 32] and are published in the grave catalogue [29]. A combination of methods was used to provide an age range for each individual, including the development of the dentition [33], and long bones metrics [34]. Based on these data, several further anthropological and archaeological studies on the population have been carried out [e.g. 35–38].

Out of 757 individuals buried, 278 (36.7%) belonged to children having died before the age of 12 years corresponding to two previously established age groups: *Infans I*: 0–6 and *Infans II*: 7–12 years old at death [37]. We selected 75 individuals from these two age groups, 34 from *Infans I* and 41 from *Infans II*. The mortality pattern for this site is U-shaped or non-catastrophic, our sample is therefore understood to represent children having died before reaching adulthood under the "normal" living conditions at the time and over a period of several centuries, and not as the result of a single event, crisis or disaster. Selection for this study was based on the preservation of the dentition and includes individuals from across the spatial distribution of the cemetery, however individuals buried with bronze artefacts are generally better preserved, perhaps due to the antibacterial properties of copper [39]; therefore, the sample may disproportionately represent children of higher status within the community.

Sex estimation using amelogenin peptides in human dental enamel is currently gaining momentum [26, 27, 40–42]. We applied nanoflow liquid chromatography-tandem mass spectrometry (nanoLC-MS/MS) to identify sex-specific peptides in the teeth of the same 75 children from Franzhausen I selected for this study [25], using a previously published protocol [28]. This procedure returned a reliable sex identification in 70 individuals and confirmed that the sex of the children corresponds to the gendered burial position in 98.4% of cases. We therefore used the gendered burial position as a proxy for sex for the remaining individuals (Fig 1).

## Palaeopathological conditions of the oral cavity and linear enamel hypoplasia

Dental caries, dental calculus, and linear enamel hypoplasia (LEH) were identified visually and by probing based on established criteria [43] and scored as present or absent. All teeth (both deciduous and permanent) were considered for a total of 978 teeth. Due to staining and taphonomic alteration of dental surface, only dental caries sufficiently advanced to form a cavitation were identified with the help of a dental explorer. Calculus was identified on the lingual and buccal surfaces. Calculus prevalence in our sample is expected to be under-estimated due to sample exposure to taphonomic erosive processes and cleaning. LEH was identified visually and by running a probe across the enamel surface to find a series of grooves generally observed on multiple teeth.

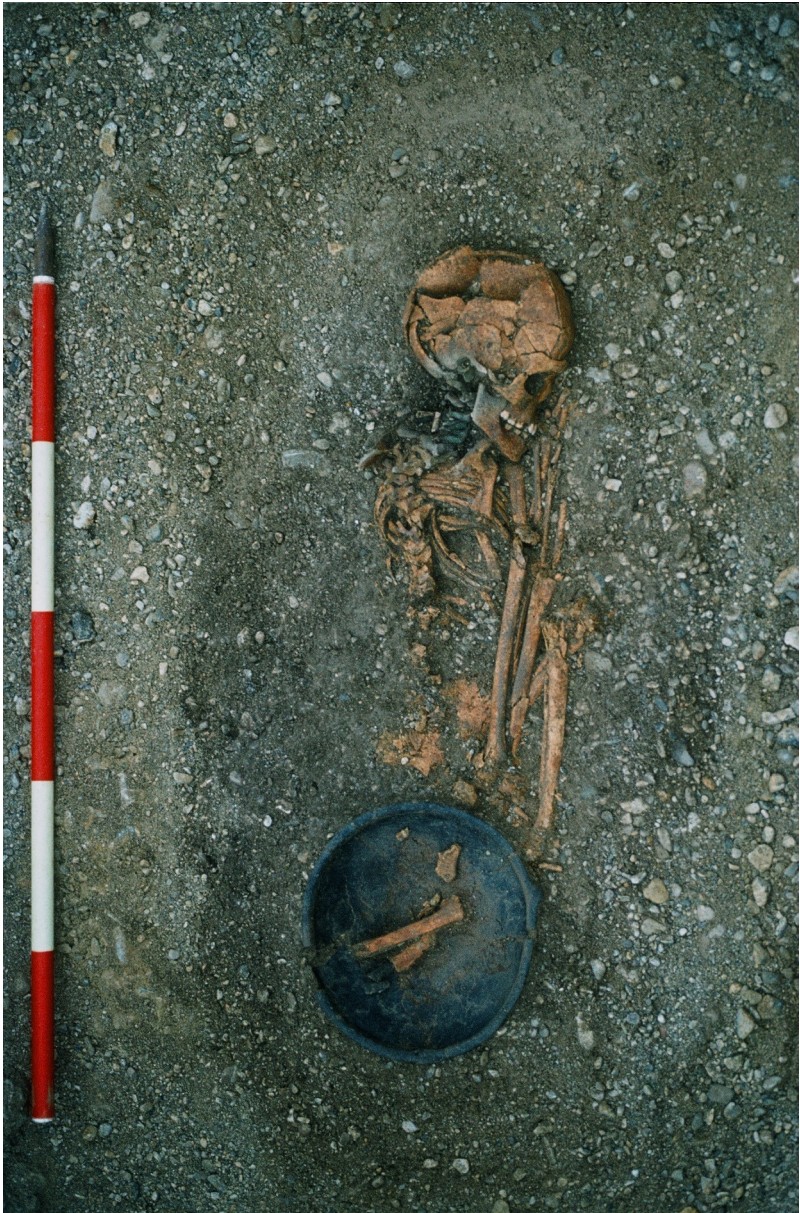

**Fig 1. Burial of a 7-9-year old boy at Franzhausen I, Austria (Verf.** 290) in flexed body position typical for males (left body side, north-south orientation) © Bundesdenkmalamt Wien/NÖ.

### Dentine exposure

Dentine exposure (DE) (Fig 2A) on the occlusal surface can be observed visually as dentine has a distinct appearance to enamel (darker and less shiny, sometimes stained green by bronze artefacts). Only deciduous molars were considered, the upper first molars (Udm1), the upper second molars (Udm2), the lower first molars (Ldm1) and the lower second molars (Ldm2). The percentage of the occlusal surface with exposed dentine was quantified following a standard procedure outlined in previous studies [16, 19, 44, 45]. A top-down photo of the occlusal surfaces was taken using a digital camera (Canon EOS 5D MARK IV) at a resolution

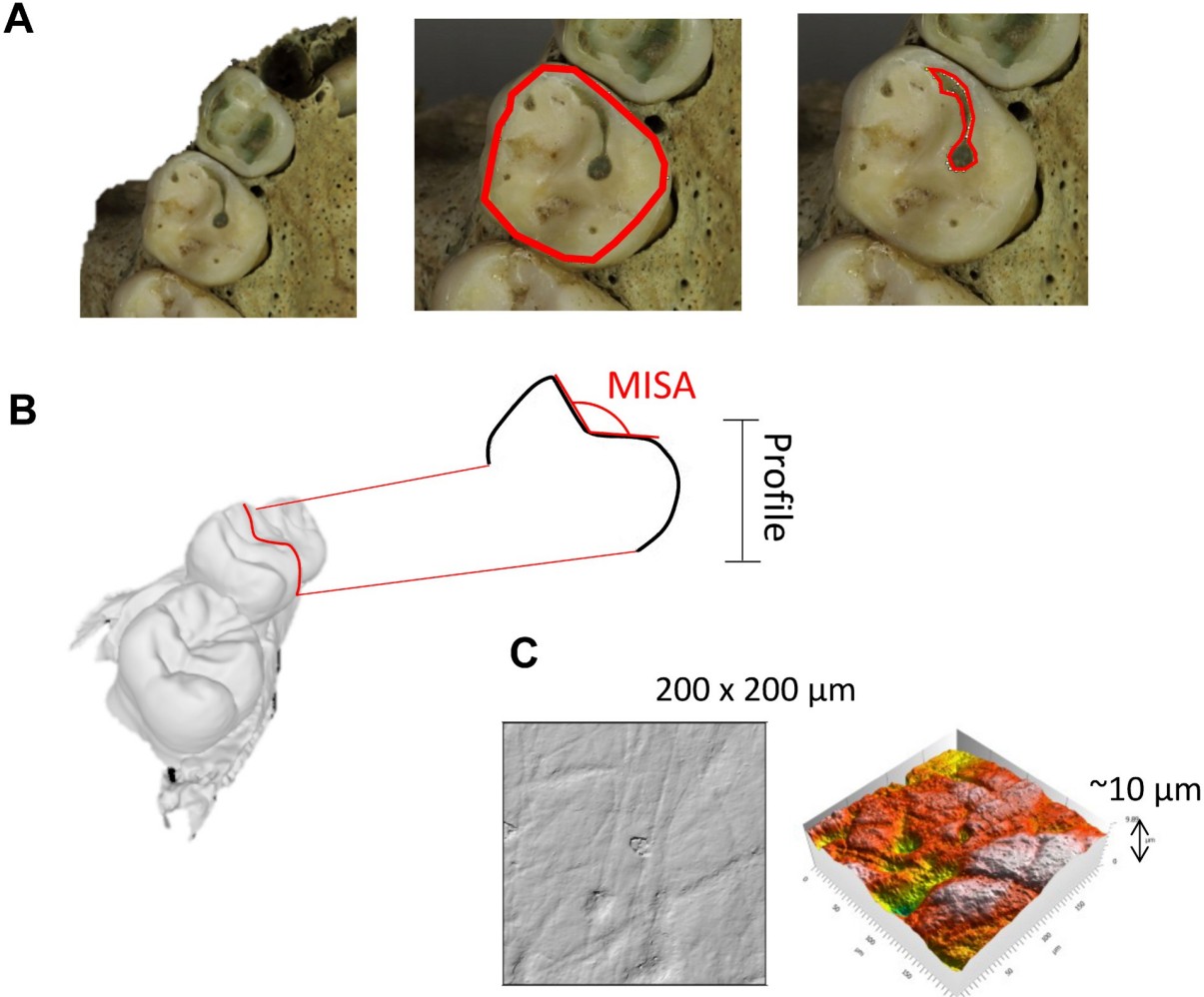

**Fig 2. Dental wear measurement methods used.** (A) Quantification of dentine exposure, left overview of dentition in occlusal view, centre occlusal area selection for measurement, right area selection and measurement of a dentine patch. (B) Mesial interior slope angle (MISA) measurement on a 2D profile extracted from a 3D model of the occlusal surface. (C) Dental microwear texture analysis using a confocal microscope, reconstructions of a 200 × 200 µm area of the enamel surface in 2D and 3D.

6720 × 4480, 72 dpi. Images were first processed using Digital Photo Professional 4 to remove any lens distortion. Then images were opened in ImageJ (FIJI) 1.53c [46], where a series of manual area measurements were made using the polygon tool to trace area outlines, the first measurement for each molar corresponded to the area of the occlusal surface, the subsequent measurements to area of exposed dentine on the surface. A python script was then used to automatically calculate the percentage of exposed dentine for each molar from the series of measurements.

## Dental occlusal topography

The mesial interior slope angle (MISA)—the angle between the two mesial cusps (Fig 2B)—of the upper second deciduous molar (Udm2) was used as a dentine independent estimate of dental wear progression for 21 individuals (based on sample and scanner availability). An STL tessellation model of each dentition was produced using a Planmeca Emerald intraoral scanner and Planmeca Romexis 5.3.2.13 acquisition software. Models were processed in Mesh Lab

v2016.12 [47] following the methodology of Bas and colleagues [48] to extract a 2D profile running through the mesial cusp tips. The angle was then manually measured in ImageJ (FIJI) 1.53c using the angle tool.

## Dental microwear texture analysis

Occlusal dental microwear texture analysis (DMTA) using a confocal microscope and scale sensitive fractal analysis (SSFA) (Fig 2C) was carried out using the standard approach outlined in previous publications [21, 49–51]. To increase sample size, either the upper second deciduous or first permanent molar was used as microwear formation is understood to be comparable between the two [21]. Teeth were cleaned using cotton soaked in water or acetone. Three impressions of the occlusal surface were made using President MicroSystemTM (Coltene) regular body polyvinylsiloxane. The first two impressions were used to remove impurities and were discarded, and measurements were taken from the third impression. 3D images of the microscopic surface over an area of 333.21 × 250.78 μm were made using a Sensofar S Neox confocal microscope piloted by SensoSCAN 6.2 acquisition software. Surfaces were then processed in Digital Surf surface imaging and metrology software (Mountains Map 8), where a custom macro was applied to prepare the surface (removing outlier points/measurement errors) and extract a 200 × 200 μm surface for measurement. Two surface texture parameters were calculated, complexity (Asfc) and anisotropy (epLsar), using Toothfrax. Complexity (Asfc) indicates the presence of features (pits, scratches) on the surface as observed at multiple scales which generally increases when opposing teeth crush food vertically during the masticatory phase two. Complexity increases with the consumption of hard or polyhedral (non-flat) foodstuffs. Anisotropy on the other hand refers to the alignment of features (scratches) and generally results from opposing teeth grinding and shearing food in repetitive jaw movements during masticatory phase two. Anisotropy increases with the consumption of soft, tough or flat foodstuffs [50–52].

## Statistical analysis and data visualization

Statistical analyses and data visualizations were carried out in R 4.0.3 [53]. Confidence intervals for prevalences were calculated using binomial tests performed with the binom.test function. Prevalences are provided both as crude prevalence rates (CPR–the proportion of individuals affected by the condition) and true prevalence rates (TPR–the proportion of observable elements, in this case teeth, affected by the condition), when relevant. Continuous variables were compared between groups using non-parametric Wilcoxon-Mann-U tests using the function wilcoxon.test to avoid assumptions of the normal distribution of variables within our samples. The relationship between variables were modelled with a linear model and linear correlation tested by Pearson's test. Data visualization was carried out using the ggplot2 R package.

## Results

### 1. Paleopathological conditions of the oral cavity and linear enamel hypoplasias

The prevalences of caries, calculus and linear enamel hypoplasias among the children are summarized in Table 1.

Dental caries were observed in only 4 individuals out of 75. Of these 4 individuals only one presented tooth decay on multiple teeth. Only 5 teeth out the 978 considered by the study were found to be carious, less than 1% of teeth. Almost all carious teeth belonged to *Infans II*

**Table 1. Crude prevalence rate (CPR) and true prevalence rate (TPR) of caries, calculus and linear enamel hypoplasia (LEH) separated by age group.**

|  | individuals | cases | CPR (%)—CI* | teeth | teeth affected | TPR (%)—CI* |
|---|---|---|---|---|---|---|
| | | | Caries | | | |
| All ages | 75 | 4 | 5% (CI 1% - 13%) | 978 | 5 | <1% (CI 0% - 1%) |
| *Infans I* | 34 | 1 | 3% (CI 0% - 15%) | 415 | 1 | <1% (CI 0% - 1%) |
| *Infans II* | 41 | 3 | 7% (CI 2% - 20%) | 563 | 4 | <1% (CI 0% - 2%) |
| | | | Calculus | | | |
| All ages | 75 | 11 | 15% (CI 8% - 25%) | | | |
| *Infans I* | 34 | 3 | 9% (CI 2% - 24%) | | | |
| *Infans II* | 41 | 8 | 20% (CI 9% - 35%) | | | |
| | | | LEH | | | |
| All ages | 75 | 5 | 7% (CI 2% - 15%) | | | |
| *Infans I* | 75 | 1 | 1% (CI 0% - 7%) | | | |
| *Infans II* | 75 | 4 | 5% (CI 1% - 13%) | | | |

* 95% confidence interval

children, over the age of 6 years. Observed caries were generally small and occurred on the occlusal or interproximal surface of deciduous and permanent molars.

Dental calculus formation was observed across the molars and incisors of 11 out of 75 children, less than a fifth, a majority of which were also over the age of 6 years.

Linear enamel hypoplasia was also rare in the sample, observed in 5 of the 75 children, less than 10%, on the apical portion of developing or developed permanent incisors, canines and in two cases premolars.

## 2. Dental wear

The prevalence of dentine exposure among deciduous molars is summarized in **Table 2**.

**Table 2. Prevalence of dentine exposure for each deciduous molar.**

|  | Teeth | Teeth affected | Prevalence (%)—CI* |
|---|---|---|---|
| | Udm1—Dentine exposure | | |
| All ages | 36 | 31 | 86% (CI 71% - 95%) |
| *Infans I* | 20 | 15 | 75% (CI 51% - 91%) |
| *Infans II* | 16 | 16 | 100% (CI 79% - 100%) |
| | Udm2- Dentine exposure | | |
| All ages | 56 | 39 | 73% (CI 56% - 81%) |
| *Infans I* | 28 | 14 | 50% (CI 31% - 69%) |
| *Infans II* | 28 | 27 | 96% (CI 82% - 100%) |
| | Ldm1- Dentine exposure | | |
| All ages | 41 | 39 | 95% (CI 83% - 99%) |
| *Infans I* | 22 | 20 | 91% (CI 71% - 99%) |
| *Infans II* | 19 | 19 | 100% (CI 82% - 100%) |
| | Ldm2- Dentine exposure | | |
| All ages | 56 | 48 | 86% (CI 74% - 94%) |
| *Infans I* | 26 | 18 | 69% (CI 46% - 86%) |
| *Infans II* | 30 | 30 | 100% (CI 88% - 99%) |

* 95% confidence interval

Over half of the young children's (*Infans I*) molars present exposed dentine. With the exception of a single upper second deciduous molar, all molars present exposed dentine among older children (*Infans II*). Among younger children, the lower first deciduous molar most commonly shows exposed dentine followed by the upper first deciduous molar. The upper second deciduous molar is the tooth that most commonly does not show any exposed dentine. Dentine exposure on the second deciduous molars is significantly more likely among older children (based on 95% confidence intervals).

With respect to the quantified proportion of exposed dentine on the occlusal surface, we observe a greater difference in dentine exposure between the first and second upper molars than between the lower molars. The rate of dentine exposure appears highest in Udm1, similar in Ldm1 and Ldm2, and lowest in Udm2 (Fig 3). For all four deciduous molars dentine exposure has a strong positive linear relationship with mean estimated age (Pearson coefficient: Udm1 0.69, Udm2 0.66, Ldm1 0.50, Ldm2 0.72). However, the mean estimated age explains only between 25% and 52% of the variation in dentine exposure depending on the molar (Linear model $R^2$: Udm1 0.48, Udm2 0.43, Ldm1 0.25, Ldm2 0.52) (Fig 3B–3E).

Measurement values for the Udm2 mesial interior slope angle (MISA) in degrees are summarized in Table 3. The angle among unworn teeth is between 90˚ and 100˚, as during *Infans I* over a period of three to four years Udm2 MISA increases to about 130˚, up to 153˚. During *Infans II* this increase continues reaching up to 166˚.

MISA measurements indicate that changes to the occlusal surface are occurring at different rates within the population, with some children showing a MISA angle 20˚ wider than some *Infans II*. Udm2 MISA has a strong positive linear correlation with mean estimated age, with 57% of the variation of MISA explained by age (Pearson coefficient: 0.75, $R^2 = 0.57$) (Fig 4A). Udm2 MISA also has strong positive linear correlation with Udm2 dentine exposure (Pearson correlation: 0.71) (Fig 4B).

Measurement of dental microwear texture is summarized in Table 4.

Microwear texture analysis suggests a generally high complexity (Asfc above 2) and low anisotropy (epLsar below 0.0020) profile for both age groups with extensive variation in complexity among sampled individuals. Complexity (Asfc) and anisotropy (epLsar) have a strong negative linear correlation (Pearson coefficient: -0.62) but are only moderately predictive of each other ($R^2 = 0.39$) (Fig 5A). Complexity is moderately negatively correlated, and anisotropy weakly correlated with mean estimated age (Pearson coefficient: Asfc -0.43, epLsar -0.14). 19% of Asfc and 2% of epLsar can be explained by mean estimated age (Asfc $R^2 = 0.19$, epLsar $R^2 = 0.02$). The youngest child between 4 and 5 years old shows a high complexity and low anisotropy. Four children between the ages of 4 and 9 years old show high anisotropy, and the three children between 7 and 9 years old exhibit low complexity. Three of the oldest children between 8 and 12 show low anisotropy and higher complexity. The general trend among children above the age of 5 years is of decreasing anisotropy around the age of 8 or 9 years (Fig 5B, 5C).

## 3. Intra-population male—female comparisons

Differences in the prevalence of caries, calculus, and dentine exposure (by tooth position) between male and female children are summarized in Table 5.

Exposed dentine is more prevalent among female children for all deciduous molars. However, 95% confidence intervals indicate that these differences are uncertain given the sample size (the 95% CI of prevalence for each variable overlaps between the sexes).

Differences in measurements of dentine exposure, occlusal topography and dental microwear texture are summarized in Table 6.

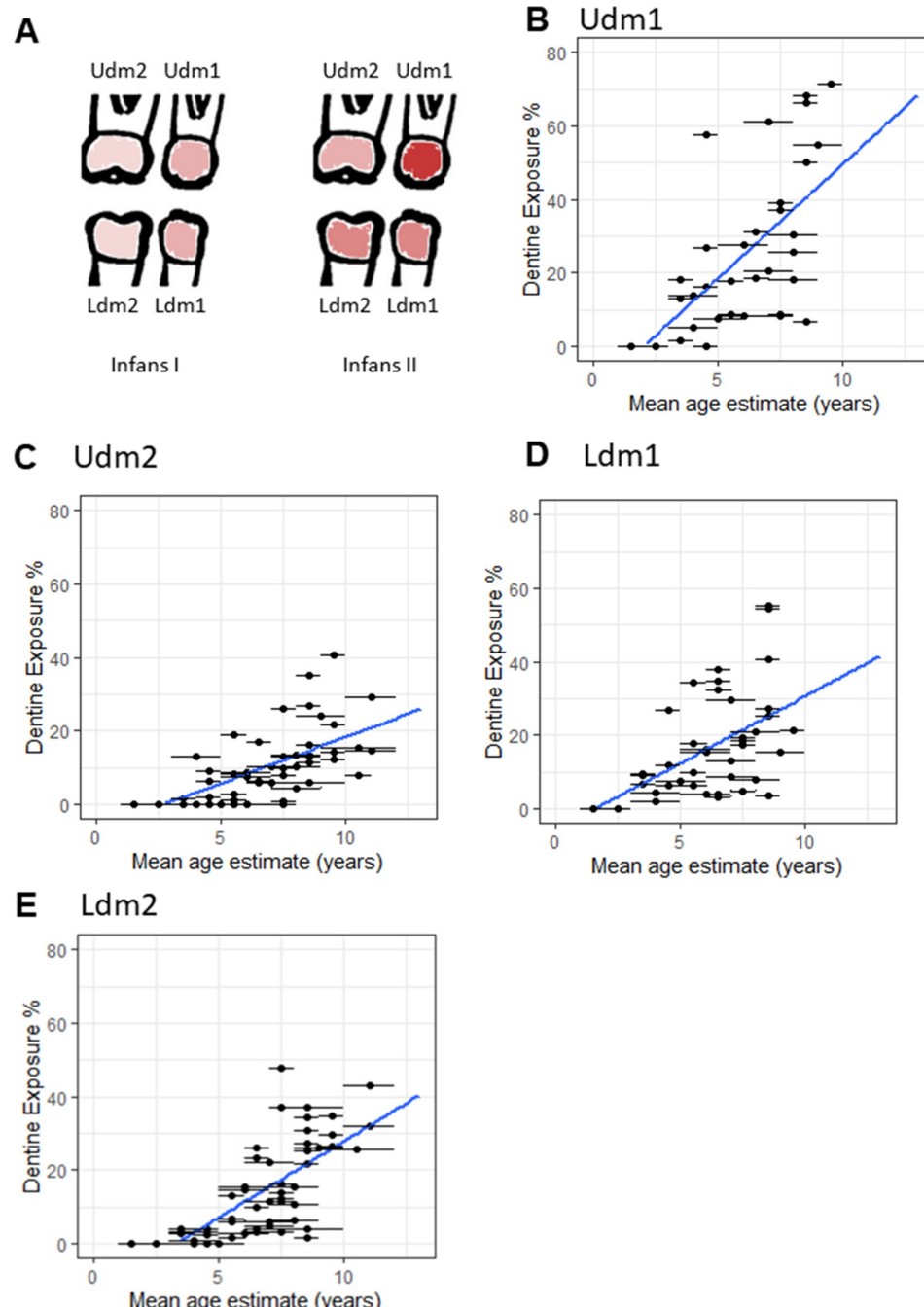

**Fig 3. Dentine exposure on deciduous molars varies depending on the position of the molar.** (A) Illustration of the distribution of dentine exposure among deciduous molars (darker red signifies a higher mean percent (%) of exposed dentine on the occlusal surface). (B) Scatterplot of Udm1 the area of dentine exposure as a % of the occlusal surface by mean estimated age with a linear model. (C) Scatterplot of Udm2 the area of dentine exposure as a % of the occlusal surface by mean estimated age with a linear model. (D) Scatterplot of Ldm1 the area of dentine exposure as a % of the occlusal surface by mean estimated age with a linear model. (E) Scatterplot of Ldm2 the area of dentine exposure as a % of the occlusal surface by mean estimated age with a linear model.

**Table 3. Upper second deciduous molar occlusal topography (Mesial interior slope angle—(MISA) in degrees).**

|  | n | Mean | SD | Minimum | Maximum |
|---|---|---|---|---|---|
| All ages | 21 | 141 | 17.6 | 90 | 166 |
| *Infans I* | 12 | 134 | 18.2 | 90 | 153 |
| *Infans II* | 9 | 151 | 11.7 | 136 | 166 |

No significant differences were identified between male and female children for dentine exposure, occlusal topography, and microwear texture, when all ages are considered together. When *Infans I* and *Infans II* age groups are considered separately, dentine exposure is significantly higher in *Infans I* female lower molars (Wilcoxon test between sexes: Ldm1 p = 0.023, Ldm2 p = 0.008) (Fig 6A–6D). Udm2 MISA measurements are similar between both sexes, with some older mostly female children showing smaller than average topographic changes due to wear (Fig 6E). The two microwear measurements show variation between sexes (Fig 6F), however, these differences remain uncertain as they may also be explained by differences in age (Fig 6G).

## Discussion

*Caries*: Dental caries are found in only a small proportion of children (4 out of 75–5%) and teeth (5 out of 978 –less than 1%) at Franzhausen I, despite the lack of modern dentistry and oral hygiene standards. This compares well with existing data on another sample from the same site, in which with caries affected only 8 out of 173 (a little under 5%) children and 10 molars and 1 canine in a sample of 1606 teeth [32]. Dental caries remain uncommon even among children with well-preserved dentitions within the sample, with only one case of multiple caries within a single dentition observed. For comparison, among industrialized societies today a broad survey of the prevalence of childhood caries in Europe from the 1990s found a prevalence of caries in primary teeth for children 5 to 7 typically ranging from 40% to 70%, occasionally as high as 87% in Slovenia in the late 1980s [54] and 40% among 6 to 7 year old

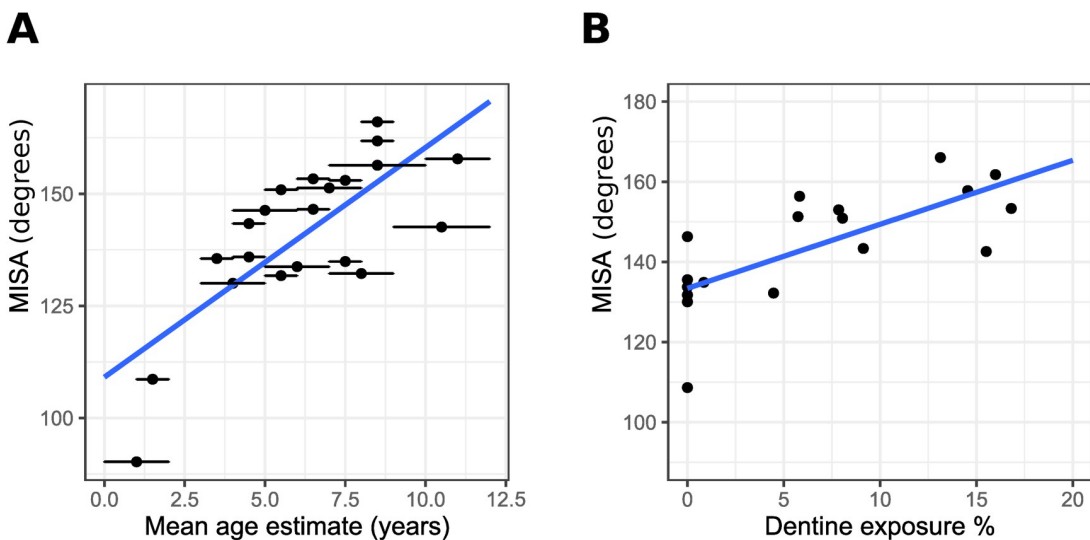

**Fig 4. Occlusal topography (MISA) also indicates both age dependent and independent intra-population dental wear variation.** (A) Scatterplot of MISA in degrees by mean estimated age with age estimation uncertainty error bars. (B) Scatter plot of MISA in degrees by dentine exposure %.

**Table 4. Upper second deciduous and first permanent molar facet nine microwear texture.**

| | n | Complexity (Asfc) | | Anisotropy (epLsar) | |
|---|---|---|---|---|---|
| | | Mean | SD | Mean | SD |
| All ages | 8 | 4 | 2.95 | 0.0019 | 0.0006 |
| *Infans I* | 2 | 7.05 | 2.82 | 0.0019 | 0.0009 |
| *Infans II* | 6 | 2.04 | 2.37 | 0.002 | 0.0006 |

children in Germany in 2015 [55]. Among pre-industrial European populations the prevalence of caries among children is generally found to be much lower, a study of childhood caries in both Early Iron-Age and Medieval Ukraine found caries in around 2% for Early Iron-Age children and 8% for Medieval children with caries affecting less than 1% of teeth [56]. Another study of Early Medieval children from multiple sites in Central Europe also found that caries affected around 1% of teeth examined [57]. A quick survey of a sample from the nearby medieval cemetery of Sankt Pölten located in the same valley as Franzhausen I found 39 out of 92 children– 42% experienced caries, with many individuals experiencing multiple caries across the dentition, however true prevalence rates have yet to be established for this sample. The prevalence of dental caries among the children of Franzhausen I is therefore low but not uncommon among pre-industrial European populations, and is indicative of a low sugar diet during childhood with few cariogenic elements. It is also possible that occlusal dental wear caused by abrasives found in food contributed by wearing away occlusal caries faster than they could develop [58]. However, the relationship between dental wear and caries remains controversial, with some studies suggesting instead that wear when resulting in dentine exposure increases the risk of caries [59, 60].

*Calculus*: Dental calculus formed among 8% to 25% of children in our sample, this figure is possibly an underestimation due to sample preservation but nonetheless reflects the limited extent of available oral hygiene practices with calculus being more prevalent than in developed countries today and in a similar range to parts of developing countries like India in the early 2000s [61, 62]. This suggests that diet and perhaps occlusal dental wear rather than oral hygiene played the most important role in the prevention of caries. However, the inverse relationship between the mineralization of dental calculus and demineralization of caries [63, 64]

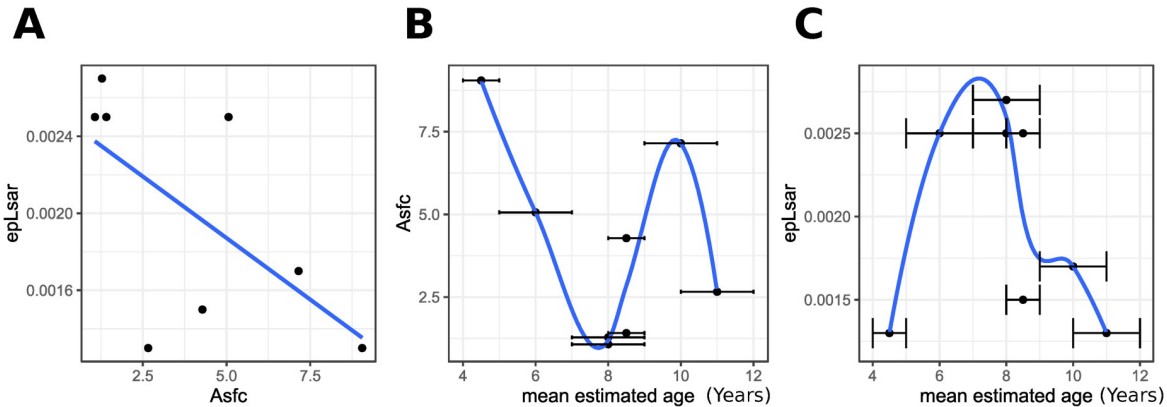

**Fig 5. Dental microwear analysis also indicates both age dependent and independent intra-population dental wear variation.** (A) Scatterplot of surface anisotropy (epLsar) by surface complexity (Asfc). (B) Scatterplot of surface complexity (Asfc) by mean estimated age with age estimation uncertainty error bars, a loess curve is used to better represent the non-linear relationship between the variables. (C) Scatterplot of surface anisotropy (epLsar) by mean estimated age with age estimation uncertainty error bars, a loess curve is used to better represent the non-linear relationship between the variables.

**Table 5. Comparison of crude prevalence rate (CPR) and true prevalence rate (TPR) of caries, calculus and deciduous molar dentine exposure between male and female children.**

| | individuals | cases | CPR (%)—CI* | teeth | teeth affected | TPR (%)—CI* |
|---|---|---|---|---|---|---|
| | | | Caries | | | |
| female | 34 | 2 | 6% (CI 1% - 20%) | 434 | 2 | <1% (CI 0% - 2%) |
| male | 38 | 2 | 5% (CI 1% - 18%) | 501 | 3 | <1% (CI 0% - 2%) |
| | | | Calculus | | | |
| female | 34 | 5 | 9% (CI 2% - 24%) | | | |
| male | 38 | 6 | 20% (CI 9% - 35%) | | | |
| | | | LEH | | | |
| female | 34 | 1 | 3% (CI 0% - 15%) | | | |
| male | 38 | 4 | 11% (CI 3% - 25%) | | | |
| | | | Udm1—Dentine exposure | | | |
| female | | | | 16 | 15 | 94% (CI 70% - 100%) |
| male | | | | 18 | 14 | 78% (CI 52% - 94%) |
| | | | Udm2- Dentine exposure | | | |
| female | | | | 24 | 20 | 83% (CI 63% - 95%) |
| male | | | | 29 | 19 | 66% (CI 46% - 82%) |
| | | | Ldm1- Dentine exposure | | | |
| female | | | | 19 | 19 | 100% (CI 82% - 100%) |
| male | | | | 20 | 18 | 90% (CI 68% - 99%) |
| | | | Ldm2- Dentine exposure | | | |
| female | | | | 26 | 25 | 96% (CI 80% - 100%) |
| male | | | | 27 | 21 | 78% (CI 58% - 91%) |

* 95% confidence interval

may also partially explain the disproportionate rate of dental caries versus calculus among the Franzhausen I children.

*Linear enamel hypoplasia*: With a prevalence between 2% and 15%, linear enamel hypoplasia (LEH) can be considered uncommon among the Franzhausen I children by historical standards [65–68]. This low LEH prevalence alone only provides limited insight into experienced stress events and childhood living conditions more broadly, however, when considered alongside the quality of child grave goods it provides further indication that the individuals buried at Franzhausen I and perhaps in particular those sample in this study came from a relatively wealthy and stable community.

*Macrowear analysis*: The prevalence of dentine exposure is high and varies somewhat depending on molar position, with most molars exhibiting prevalence between about 70% and 100%, with exception of the Udm2 with a lower prevalence of only 56% to 81%. This variation between tooth positions can in part be explained by differing eruption times between first, second, lower, and upper deciduous molars, but also their morphology and occlusion, which influences where enamel is removed and how much needs to be removed before dentine is exposed. Indeed, lower dentine exposure in Udm2 is expected as Udm2 erupts last out of the four and is known to have a thicker enamel layer than other deciduous molars [69]. In children of present-day industrialized societies, dentine exposure generally occurs at the earliest around the age of 6 or 7 years and even among children aged 12 to 14, dentine exposure is only observed in a few children (sometimes less than 10%) [18, 70, 71]. This contrasts strongly with the children of Franzhausen I, where more than half of the *Infans I* children show exposed dentine on deciduous molars. When compared to Mays and Pett's (2014) study of medieval

**Table 6. Comparison of dentine exposure (area of exposed dentine in % of the occlusal surface), occlusal topography (MISA, mesial interior slope angle), and dental microwear texture (Asfc and epLsar) between male and female children.**

|  | n teeth | mean | SD |
|---|---|---|---|
| Udm1—Dentine exposure (%) | | | |
| female | 17 | 26.0% | 22.4% |
| male | 13 | 31.6% | 24.4% |
| Udm2—Dentine exposure (%) | | | |
| female | 24 | 9.3% | 8.2% |
| male | 22 | 9.1% | 8.0% |
| Ldm1—Dentine exposure (%) | | | |
| female | 20 | 17.3% | 13.0% |
| male | 16 | 16.7% | 13.8% |
| Ldm2—Dentine exposure (%) | | | |
| female | 26 | 13.5% | 12.6% |
| male | 22 | 15.8% | 14.1% |
| Udm2—MISA | | | |
| female | 11 | 145 | 12.7 |
| male | 7 | 134 | 25.6 |
| Udm2/M1—microwear complexity (Asfc) | | | |
| female | 3 | 2.51 | 2.21 |
| male | 5 | 4.88 | 3.19 |
| Udm2/M1—microwear anisotropy (EpLsar) | | | |
| female | 3 | 0.0025 | 0.0000 |
| male | 5 | 0.0017 | 0.0005 |

childhood dental wear, Ldm1 and Ldm2 dentine exposure is similar, with slightly more wear in the Franzhausen I population, most noticeably with more Ldm1 dentine exposure among younger children of Franzhausen I [16]. Another study of medieval children finds that more than 50% of Udm2 present exposed dentine from the age of 5 years onwards, similar for the prevalence of dentine exposure in Udm2 estimated for the *Infans I* age group as a whole (0 to 6 years old) at Franzhausen I [17]. Compared to previously published data on childhood dental wear in individuals from the medieval Sankt Pölten cemetery located in the same valley as Franzhausen I, the prevalence of *Infans I* dentine exposure for Udm2 is about 30%, a little lower than the 31% to 69% estimate for the Franzhausen I population, but the prevalence then becomes similar in older children.

For *Infans I* children MISA measurements are also on average higher for Franzhausen I (mean MISA 134˚) than Sankt Pölten (mean MISA 119˚), but similar for *Infans II* children [48].

These comparisons of dentine exposure and MISA provided above suggest that the amount of dental wear experienced by Franzhausen I children was not only high by today's standards; and, among younger children, but also a little higher than experienced by later pre-industrial children from the same region. Within the sample it also seems that some individuals accumulate dental wear faster than others. One explanation for the existence of variation between individuals in the amount of dental wear experienced after accounting for age and the uncertainty of age estimation are differences in the physical and material properties of childhood diet within the population, either at a given time, or as a result of changes in childhood diet over the period the cemetery was in use. Despite the small size (~60 concurrent individuals), dental wear would therefore indicate that diet varied substantially between children within this Early Bronze-Age community. A caveat to this interpretation however is that measured macroscopic

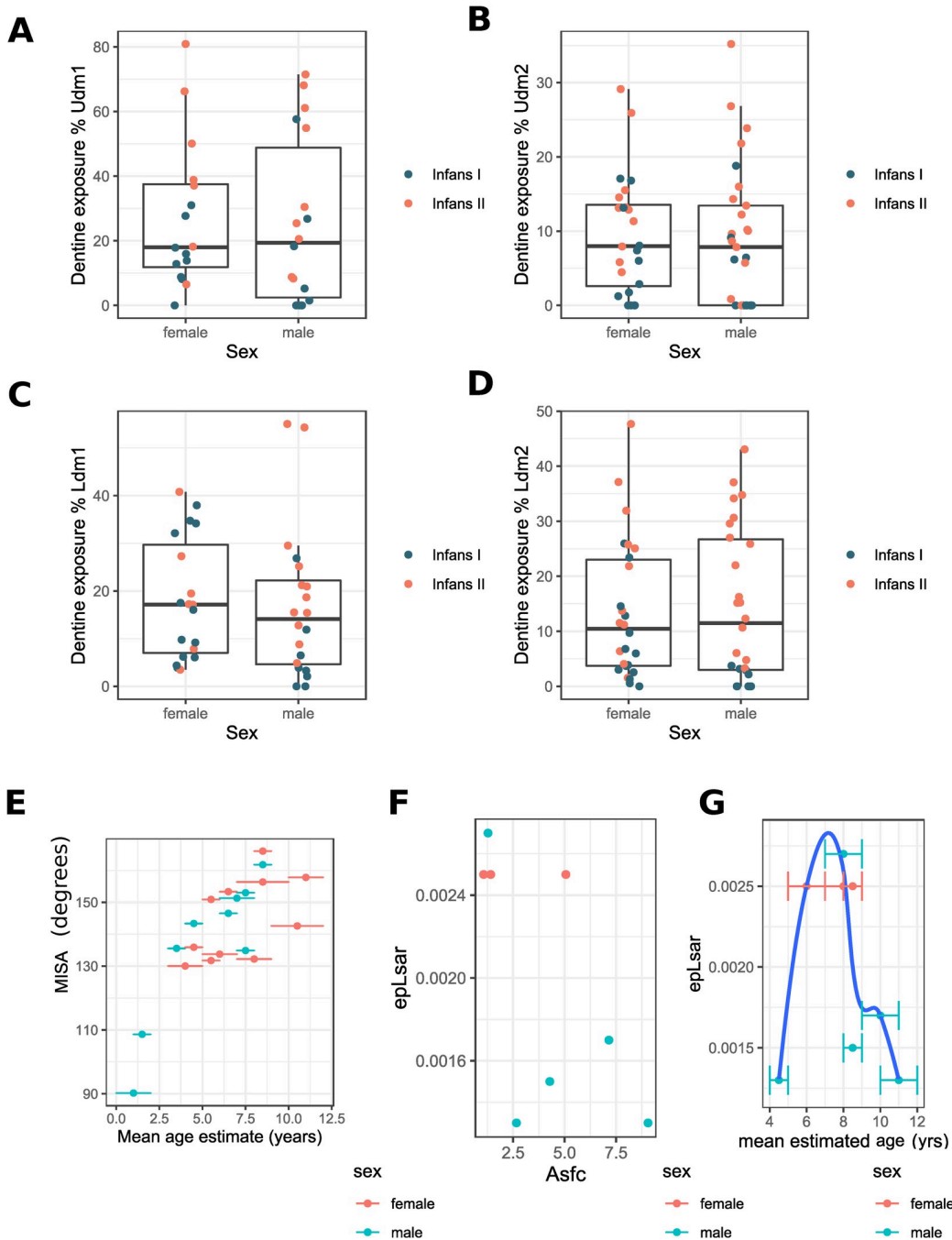

**Fig 6. Sex-based differences in dental wear.** (A) Boxplot with jitter of the area of dentine exposure as a % of the occlusal surface of Udm1 divided by sex and coloured by age group. (B) Boxplot of the area of dentine exposure as a % of the occlusal surface of Udm1 divided by sex and coloured by age group. (C) Boxplot with jitter of the area of dentine exposure as a % of the occlusal surface of Udm1 divided by sex and coloured by age group. (D) Boxplot of the area of dentine exposure as a % of the occlusal surface of Udm1 divided by sex and coloured by age group. (E) Scatterplot of MISA by mean estimated age with age estimation uncertainty error bars coloured by sex. (F) Scatterplot of surface anisotropy (epLsar) by surface complexity (Asfc) coloured by sex. (G) Scatterplot of surface anisotropy (epLsar) by mean estimated age with age estimation uncertainty error bars, a loess curve is used to better represent the non-linear relationship between the variables, coloured by sex.

dental wear variation results both from differences in environment, diet, and food preparation (discussed here), and innate differences in enamel thickness and dental tissue proportions of unworn teeth (requiring further imaging of internal structures and not considered in the scope of this study). It is therefore conceivable that some innate differences also contributed to some degree to the observed dental wear variation described here. Despite this, variation in dental wear among children within the Franzhausen I population can be observed both in terms of dentine exposure and occlusal topography, both correlated yet dependent on different underlying innate features, therefore the use of multiple measurements of dental wear lends support to the idea that some variation of childhood diet occurred within the population.

*Dental microwear analysis*: Although dental microwear analysis is limited to a small subset of 8 individuals due to taphonomy, among the individuals that could be measured, the microwear signature is generally one of high complexity (mean Asfc above 2) and low anisotropy (mean epLsar below 0.0020). The traditional interpretation of such a signature based on the physical properties of food would suggest a coarse diet with notable harder elements—either the food itself or exogenous particles consumed with food—especially among older children. Another approach to the interpretation of microwear based instead on mechanical properties of diet would suggest a diet containing a high proportion of polyhedral particles (as opposed to one dominated by "flat" foodstuffs) [52]. When compared to children from ancient Rome and Medieval England, the children of Franzhausen I have more complex and less anisotropic microwear textures, suggesting diet was harder/polyhedral and required less repetitive jaw movements [15, 21, 72]. The results suggest that children before the age of 8 or 9 may have consumed fewer hard food components commonly found outside the domestic setting than their older counterparts, similarly to the observations for medieval children made by Mahoney et al. [21]. However, bite force does increase throughout ontogeny [73] and may at least partially explain the differences in on dental microwear (and macrowear) formation between *Infans I* and *II* age groups.

*Dietary interpretation*: People buried at Franzhausen I likely cultivated a rich spectrum of crops including barley, einkorn and emmer wheats, spelt and lentils [74], and raised domesticated animals such as cattle, caprids, and pigs [75, 76]. The processing of cereals would be carried out with small grinding stones [77] and foods could be cooked in ceramic pots. Studies of childhood diet in other pre-industrial agrarian communities suggest that younger children (*Infans I*) would have consumed a relatively soft, semi-liquid diet of gruels, paps, or broths [21, 22, 78], and older children from the age of five to six more solid foods with a diet closer to that of adults [22]. Based on our analysis of oral pathology and dental wear, we suggest that in general rather than softer chewy and cariogenic, or sugary foods like porridges, soft fruits, and honey, the children of Franzhausen I consumed a coarser and low sugar diet/low cariogenic meals such as vegetable, lentil and meat broths and stews, with the regular inclusion of more polyhedral foodstuffs especially among older children such as dried foods, meats, hazelnuts, chestnuts, acorns, and likely harder foodstuffs such as wild root vegetables. We also suggest that this diet could vary substantially between children.

*Sex-based differences*: Variations in dental wear between male and female children are minor, but there is a slightly higher rate of dentine exposure among younger female children, especially on the lower molars. The exact causes of this discrepancy is unclear, we can only suggest some possible explanations that require further investigation. More wear accumulating on the teeth that erupt first could be explained by differences in diet during early childhood between male and female children, with female children consuming a more abrasive diet (including more abrasives likely found in ground cereals and/or requiring more chewing); alternatively, the weaning process might have started earlier for female children with solid foods being introduced earlier on and becoming more common by the time the molars erupt.

Follow up studies of nitrogen isotopes comparing male and female children could corroborate this hypothesis. Dental microwear does not provide clear information on dietary differences between the sexes, as the well-preserved teeth from a few individuals vary simultaneously in both sex and age. If observed differences were caused by dietary differences between the sexes and not due to age as previously suggested, they would indicate that female children consumed a softer diet with more repetitive jaw movements (i.e. surface would present higher anisotropy and lower complexity values), consuming perhaps fewer hard or unprocessed foodstuffs that are often sourced away from the domestic environment.

## Conclusion

This study of 75 non-adult dentitions from Early Bronze-Age Franzhausen I found that the estimated prevalence of caries, calculus, and exposed dentine, as well as the quantitative measurement of occlusal dentine exposure, topography (MISA), and microwear texture (Asfc, epLsar) are indicative of a significantly less cariogenic, coarser, and more abrasive diet than consumed by children today. The properties of this Early Bronze Age diet were found to be similar, but a little more coarse and abrasive than the diet of children in later pre-industrial (medieval) European populations. This result fits with the general trends observed in adults of decreasing dental wear since the Palaeolithic, up until the development of modern dentistry. The non-pathological nature of observed dental wear and the low prevalence of caries suggest relatively good oral health during childhood at Franzhausen I, and the low prevalence of LEH reflects positively on childhood health in general (despite a high childhood mortality by modern standards). Age independent dental wear variation suggests that childhood diet likely changed over the centuries the cemetery was in use, and possibly differed between children within the community at any given time. Results also highlight the possibility of modest dietary differences during childhood between younger children of different sexes. However, limitations of our dental wear analysis highlight the need for more information on the variation of internal dental structure in past populations, to further untangle the genetic and environmental factors behind macroscopic dental wear variation. Further studies of non-adult dentitions from skeletal remains, both pre- and post-Neolithic and pre- and post-industrial, will confirm or question the assumed general trends and our understanding of the co-evolution of food preparation technologies and the human dentition, as well as provide a clearer picture of the past diversity of childhood dental wear and oral pathology.

## Supporting information

**S1 Table. Samples, observations and raw measurement data.** This table contains the data on identity, pathology, and dental wear used in the study. All dental wear measurements and the individuals they are associated with are included.
(XLSX)

## Acknowledgments

We thank the staff of the Department of Anthropology at the Natural History Museum in Vienna, Karin Wiltschke-Schrotta, Margit Berner and Sabine Eggers, for their support and for granting access to the human remains under their curation.

We thank Lukas Janker, Patricia Bortel, and Christopher Gerner for their analysis of amelogenin peptides.

The Federal Monuments Authority Austria, Department Archaeology, has granted reproduction permissions for the excavation photographs.

## Author Contributions

**Conceptualization:** Marlon Bas, Katharina Rebay-Salisbury, Fabian Kanz.

**Data curation:** Marlon Bas.

**Formal analysis:** Marlon Bas.

**Funding acquisition:** Katharina Rebay-Salisbury.

**Investigation:** Marlon Bas.

**Methodology:** Marlon Bas, Christoph Kurzmann, John Willman, Fabian Kanz.

**Project administration:** Fabian Kanz.

**Resources:** Christoph Kurzmann, Doris Pany-Kucera, Katharina Rebay-Salisbury, Fabian Kanz.

**Software:** Christoph Kurzmann.

**Supervision:** Fabian Kanz.

**Validation:** John Willman.

**Visualization:** Marlon Bas.

**Writing – original draft:** Marlon Bas.

**Writing – review & editing:** Christoph Kurzmann, John Willman, Katharina Rebay-Salisbury, Fabian Kanz.

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
