## [Decision Letter · Decision Letter 0]

4 Sep 2022

PONE-D-22-19889Deciduous dental pathology and wear among sex determined Early Bronze-Age children from Franzhausen I, Lower AustriaPLOS ONE

Dear Dr. Kanz,

Thank you for submitting your manuscript to PLOS ONE. After careful consideration, we feel that it has merit but does not fully meet PLOS ONE’s publication criteria as it currently stands. Therefore, we invite you to submit a revised version of the manuscript that addresses the points raised during the review process.

The manuscript by Bas and coauthors presents and discusses some oral pathologies in a sample of individuals aged inferiors to 12 years from a Bronze Age necropolis in Austria.

The odontoskeletal sample was the subject of previous morphological analysis, and for the same individuals presented in the manuscript, sex was diagnosed by proteomic analysis of dental enamel and already published in a previous article.

Thus, this paper aims to present the prevalence (but see reviewer's comments 2) of some oral diseases divided by sex.

The manuscript has probably already been submitted to another journal, as is evident from the structure of the abstract and improper bibliographic formatting. In addition, the authors failed to put either page numbers or line numbers in the manuscript, thus making the reviewers' work more difficult.

The two reviewers made substantial criticisms that need to be considered by the authors, especially at the level of comparison with more recent populations.

In general, the authors tend present data of interest but reach conclusions that are sometimes speculative and not well supported by the results.

What emerges from the manuscript is that there is no significant difference in dental disease and wear between male and female infants and children and that patterns of wear and caries differ from those in later populations. Discussion and conclusion instead are complex and seldom provide interpretations that are not supported by clear and indisputable trends in the data .

Particularly speculative are the conclusions regarding the differential onset of weaning between males and females, and generally the topic of weaning is treated too easily and without an accurate understanding (see Humphrey 2014) of the complexity of the phenomenon, which in any case cannot be approached in terms of dental wear.

To these general criticisms there are other problems in the manuscript that need to be addressed by the authors:

“Many of these deaths were likely due to accidents or rapid infections and not chronic disorders, so the sample is generally representative of children within the community.”

As stressed by reviewer 1 this sentence has no support in the manuscript. In general the mortality nature of the sample is not enough discussed by the authors.

Linear enamel hypoplasia (which is rare and insignificant in the sample) is the outcome of other pathological of physiological processes during development and shouldn’t in my opinion clustered among oral pathologies.“Contrary to a prior publication (38), linear enamel hypoplasia could be observed in only 5 of the 75 children, a prevalence of about 7 % (95 % CI between 2 % and 15 %).” It is necessary to explain why the two analyses on the same dental series can produce different results. What was the percent of LEH in 38?“When plotted with error bars for age estimation uncertainty and a loess curve to illustrate the general trend, we observe that the variation in dentine exposure between individuals of a similar age is not entirely explained by age estimation uncertainty for all four molars”. I think that the use of lowess (and not loess) smoothing regression is not correct in this analysis and do not add any information about the correlation between age at death and dentine exposure. Moreover lowess curves should be plotted with the confidence interval of the locally weighted polynomial regression fit. I suggest the authors to use linear (maybe robust) regression models.“Based on existing literature, the people buried at Franzhausen I likely began weaning their children before the age of two but a little later than in industrialized populations (75, 76)” this sentence is misleading and wrongly suggests that Humphrey and/or Sellen eventually discussed data about Franzhausen. The onset of weaning is physiologically necessary before one year of age and this well known and discussed by a large literature.“Lipid analysis carried out on small ceramic vessels used for infant feeding from the Late Bronze Age and Iron Age in the region confirmed that they contained animal milk (77)” The presence of milk in ceramic vessel for feeding infants DO NOT imply they were weaned. The authors are here confusing the onset of weaning with the use of non human milk in nursing infants. The onset of weaning foresees the introduction of non milk (possibly soft and processed) food into the infant diet.“the observation that the upper and lower dm1 wear appears to start wearing shortly after eruption confirms that weaning had indeed begun by at least 2 years old.” This sentence offers no clues on the nursing in Franzhausen as the onset of weaning before 2 years old is a practice not only well known for all humans (including neanderthals).Variations in dental wear between male and female children are minor, but there is a slightly higher rate of dentine exposure among younger female children, especially on the lower molars. More wear accumulating on the teeth that erupt first could be explained by differences in diet during early childhood between male and female children, with female children consuming a more abrasive diet (including more abrasives likely found in ground cereals and/or requiring more chewing); alternatively, the weaning process might have started earlier for female children with solid foods being introduced earlier on.” The difference between sexes for dentine exposure il limited to the second lower molar (Ldm2 p = 0.008). But Ldm2s are in occlusion after age 2. Therefore, this unique and not extraordinary evidence tells nothing about differential onset of weaning. Please submit your revised manuscript by Oct 19 2022 11:59PM. If you will need more time than this to complete your revisions, please reply to this message or contact the journal office at plosone@plos.org. Please include the following items when submitting your revised manuscript:A rebuttal letter that responds to each point raised by the academic editor and reviewer(s). You should upload this letter as a separate file labeled 'Response to Reviewers'.A marked-up copy of your manuscript that highlights changes made to the original version. You should upload this as a separate file labeled 'Revised Manuscript with Track Changes'.An unmarked version of your revised paper without tracked changes. You should upload this as a separate file labeled 'Manuscript'.

We look forward to receiving your revised manuscript.

Kind regards,

Luca Bondioli, M.D.

Academic Editor

PLOS ONE

Journal Requirements:

2. In your manuscript, please provide additional information regarding the specimens used in your study. Ensure that you have reported specimen numbers and complete repository information, including museum name and geographic location. 

For more information on PLOS ONE's requirements for paleontology and archaeology research, see https://journals.plos.org/plosone/s/submission-guidelines#loc-paleontology-and-archaeology-research.

"This study was undertaken within the framework of the ERC project ‘The value of mothers to society: responses to motherhood and child rearing practices in prehistoric Europe’. This project has received funding from the European Research Council (ERC) under the European Union’s Horizon 2020 research and innovation programme (grant agreement No 676828, PI Katharina Rebay-Salisbury)."

"This study was undertaken within the framework of the ERC project ‘The value of mothers to society: responses to motherhood and child rearing practices in prehistoric Europe’. This project has received funding from the European Research Council (ERC) under the European Union’s Horizon 2020 research and innovation program (grant agreement No 676828, PI Katharina Rebay-Salisbury). The funders had no role in study design, data collection and analysis, decision to publish, or preparation of the manuscript."

"The authors declare that they have no known competing financial interests or personal relationships that could have appeared to influence the work reported in this paper."

Reviewers' comments:

Reviewer's Responses to Questions

**Comments to the Author**

1. Is the manuscript technically sound, and do the data support the conclusions?

Reviewer #1: Yes

Reviewer #2: Yes

2. Has the statistical analysis been performed appropriately and rigorously? 

Reviewer #1: Yes

Reviewer #2: I Don't Know

3. Have the authors made all data underlying the findings in their manuscript fully available?

Reviewer #1: Yes

Reviewer #2: No

4. Is the manuscript presented in an intelligible fashion and written in standard English?

Reviewer #1: Yes

Reviewer #2: Yes

5. Review Comments to the Author

Reviewer #1: The manuscript by Bas and co-authors reports on oral pathologies (caries, calculus and linear enamel hypoplasia) and macro- and micro-wear texture analyses in a sample of deciduous teeth from 75 individuals aged less than 12 years from the Early Bronze-Age necropolis of Franzhausen I in Austria. All the variables are considered in relation to age and sex of the children, to investigate childhood diet and related oral health.

The authors applied a classic methodological approach alongside with more advanced methods (amelogenin peptide analysis and microwear texture analysis). The study is well designed, and the manuscript is well structured and clearly written. The data are clearly presented and analysed.

The authors should check carefully the references: there are some missing, some duplications and some in the wrong format.

A major concern is about the statement in Materials and methods: “Many of these deaths were likely due to accidents or rapid infections and not chronic disorders, so the sample is generally representative of children within the community”. It seems to me too speculative, and I don’t see how the authors can declare this. I suggest to the authors to carefully explain and justify this statement otherwise delete it. In any case, I would add a reference to the Osteological paradox.

The authors should better explain the two surface texture parameters, i.e., complexity and anisotropy, for non-specialists readers and to better understand the discussions. A description of what they are, and the meaning of their variation is in my view necessary.

The authors must carefully check the figure numbering. There are several errors.

- In results, Dental wear, Fig. 3 B,C,D,E should be Fig. 3 C,D,E,F;

- Reference to Fig. 3F is wrong,

- Fig. 4 G,H,I,J should be Fig.4 A,B,C,D.

- Caption of Fig.4: Please note that the ranges of the x-axes are not the same, should be y-axes.

Results, Microwear texture analysis: It is hard to me to follow the consideration about the children age and anisotropy and complexity. It is not clear if the age ranges reported are wrong or follow a line of reasoning that I cannot see. Moreover, some of the “low” and “high” reported seems to not correspond to the graph. I suggest to the authors to better explain it and make the data reporting clearer also graphically, this latter to better distinguish between the different individuals.

Discussions: The authors use modern, roman, and medieval samples from previous studies to discuss the Early Bronze Age children from the present study. I agree with this because the lack of comparative data from deciduous teeth from prehistoric samples. Said this, the authors should discuss the chronology of their sample first, highlighting that the differences could be related to the different subsistence economy and to discuss in the dietary interpretation the adult prehistoric diet inferred from dental microwear (Mahoney 2016 reports that no difference are scored about microwear texture analysis between permanent and deciduous teeth).

Discussion, dietary interpretation: Based on existing literature, the people buried at Franzhausen I likely began weaning their children before the age of two but a little later than in industrialized populations.

How can the authors support this statement with the data? The method here used is too coarse in time resolution to detect this subtle variation.

Other minor comments

Introduction, second paragraph: “The first objective of this study is to document childhood macroscopic and microscopic dental wear as well as dental caries and dental calculus in non-adults buried at Franzhausen I—focused onindividuals with preserved …”, correct “onindividuals”.

Materials and methods, sample paragraph, refers to Fig.1 somewhere in the text. Reference to Fig.1 is missing in the main text.

Materials and methods, sample, last paragraph: references 41-45 include two repetitions, namely #26-27.

Table 1, caption: LEH is not reported in the table.

Table 6, Age column, first two rows report 34 and 36 (I think is the number of samples).

Reviewer #2: This manuscript aims to provide an analysis of deciduous dental pathology and wear from children excavated from a Bronze Age site in Austria. The analysis provides original data and, as there are very few studies of deciduous dental pathology and wear has some value, although pathology is relatively limited and the wear patterns not especially revealing. There are some issues with the data as it's currently presented that do need to be resolved. In particular the authors need to present the dental pathology data as true prevalence rates - as they currently seem to just be crude prevalence rates, which is inadequate. There also needs to be greater clarity as to what teeth are being recorded. For example, in a child of 8 years with mixed dentition (permanent and deciduous) are only the deciduous teeth included here? If not, then the title of the paper should be altered. Overall, there is a lack of specificity with the dental pathology.

I was confused about the point of comparing BA skeletons with present-day (which is so different in terms of sugar consumption it seems a pointless comparison) - it's also confusing when the term 'industrialized' is used here when the authors mean present-day rather than 1800s onwards. Technically Europe is 'post-industrial' now. It would have been much more appropriate to focus on comparisons with other pre-industrial societies. There is some attempt at comparison with a medieval population but the authors should consider the work of Heidi Dawson in more detail, and also the work on deciduous dental pathology by Sian Halcrow - while these are from different periods and places they are still relevant for the discussion. Overall, the fact that the children have a sex estimated and comparisons between boys and girls could be made was useful and novel for a study such as this, although differences are pretty minimal. It would be more informative to see differences between the isotopes between boys and girls than wear.

Comments

- The title states that this study focuses on the deciduous dentition. The abstract does mentions only 'children'. An age range needs be given as 'children' is a socially specific term. Later in the paper the authors refer to 'under 12s', but these children may have both deciduous and permanent teeth. There needs to be greater assurance that the dental pathologies recorded are only of deciduous teeth (which is not clear). For example, at one point in the paper the authors discuss recording wear on the first permanent molar - so clearly at times teeth other than 'deciduous' are being used. The parameters of this study are very confusing. Please clarify.

-Introduction - tell us the date of the site when first introduced

-Burials are divided into 'infans 1' and 'infans ii' but the age range of each of these categories should be included in the materials and methods text, saying how many individuals in each of the categories. Information for this is only provided later. I also wonder about the choice of these categories given that the discussion centres more on weaning, which will be obscured by the 0-6 year clustering.

-Table 1 needs redoing - provide TPRs and provide more detail on the teeth most affected. Are only deciduous teeth included in this table? If permanent are as well then this needs to be made explicit in the discussion. If only deciduous then more information is required. It's only in the discussion that the authors state that caries affected ten molars and one canine in a sample of 1606 teeth - this should be in the results. Why is the LEH not included in the table of pathologies? Further explanation is required. In terms of the intra-populations comparison of pathologies in Table 5 - this should also be TPRs.

- With the dental wear comparisons of younger and older children need to be clear that when refer to molars they mean deciduous - the language needs to be more precise.

- In the discussion don't equate industrialized societies with the 1990s in Europe. This is confusing and happens regularly.

-In the discussion the authors suggest that "The results suggest that children before the age of 8 or 9 may have consumed

fewer hard food components commonly found outside the domestic setting than their older

counterparts". Please explain more clearly what is meant by this.

- Some of the Figures (e.g. 3 - 7) look a bit blurred on my version.

6. PLOS authors have the option to publish the peer review history of their article (what does this mean?). If published, this will include your full peer review and any attached files.

Reviewer #1: No

Reviewer #2: No

---

## [Author Response · Author response to Decision Letter 0]

28 Oct 2022

Dear Dr. Bondioli and reviewers,

We would like to thank you for your excellent feedback and for taking the time to provide us with constructive criticism. We have taken the points you have raised into consideration and done our upmost to correct and improve the manuscript accordingly, as a result we find the manuscript has been greatly improved by this process and we hope you will agree. In this letter we will reply more specifically to the issues you raised to explain exactly how we addressed them, and if necessary clear up any misunderstandings. We have inserted our responses in blue directly into the reviews we received, so that our answers are easier to follow and for more minor comments we simply made the corrections directly.

Response to the editor:

The manuscript by Bas and coauthors presents and discusses some oral pathologies in a sample of individuals aged inferiors to 12 years from a Bronze Age necropolis in Austria. The odontoskeletal sample was the subject of previous morphological analysis, and for the same individuals presented in the manuscript, sex was diagnosed by proteomic analysis of dental enamel and already published in a previous article. Thus, this paper aims to present the prevalence (but see reviewer's comments 2) of some oral diseases divided by sex.

The manuscript has probably already been submitted to another journal, as is evident from the structure of the abstract and improper bibliographic formatting. In addition, the authors failed to put either page numbers or line numbers in the manuscript, thus making the reviewers' work more difficult. � Firstly, we apologize for any oversights regarding the formatting. We wish however to make perfectly clear this is the first journal to which this manuscript has been submitted. We will be more careful to take the particularities of the journal into consideration in the future. 

The two reviewers made substantial criticisms that need to be considered by the authors, especially at the level of comparison with more recent populations. In general, the authors tend present data of interest but reach conclusions that are sometimes speculative and not well supported by the results. What emerges from the manuscript is that there is no significant difference in dental disease and wear between male and female infants and children and that patterns of wear and caries differ from those in later populations. Discussion and conclusion instead are complex and seldom provide interpretations that are not supported by clear and indisputable trends in the data. � This point will be addressed below but we have taken this objection into account, we agree that it is very important to clarify what constitutes a clear and direct conclusion and what is merely a hypothesis or suggestion that invites some further exploration.

Particularly speculative are the conclusions regarding the differential onset of weaning between males and females, and generally the topic of weaning is treated too easily and without an accurate understanding (see Humphrey 2014) of the complexity of the phenomenon, which in any case cannot be approached in terms of dental wear. �The relationship between the introduction of solid foods during weaning and early dental wear is still under investigation, but is not the topic of this study. But we have tried to make our line of reasoning clearer in this section as explained below.

To these general criticisms there are other problems in the manuscript that need to be addressed by the authors:

1. “Many of these deaths were likely due to accidents or rapid infections and not chronic disorders, so the sample is generally representative of children within the community.” As stressed by reviewer 1 this sentence has no support in the manuscript. In general the mortality nature of the sample is not enough discussed by the authors. � We have added a quick discussion of the mortality pattern in the sample are corrected this statement.

2. Linear enamel hypoplasia (which is rare and insignificant in the sample) is the outcome of other pathological of physiological processes during development and shouldn’t in my opinion clustered among oral pathologies. � This is probably because we originally used the term dental pathology instead of oral pathology. We have separated LEH. However, we would argue that it precisely is because it is rare in this sample that it is in fact significant and should be mentioned.

3. “Contrary to a prior publication (38), linear enamel hypoplasia could be observed in only 5 of the 75 children, a prevalence of about 7 % (95 % CI between 2 % and 15 %).” It is necessary to explain why the two analyses on the same dental series can produce different results. What was the percent of LEH in 38? � The previous analysis was conducted on a different sample from the same population, including also many adults. As the analysis took place over 30 years ago it is not quite sure what methods were used. Out of caution we initially considered not mentioning LEH in our study, but after double checking with a separate observer we confirmed that we indeed have a very low rate of LEH. We mentioned the other study out of caution, but we have decided to remove it as it only adds confusion.

4. “When plotted with error bars for age estimation uncertainty and a loess curve to illustrate the general trend, we observe that the variation in dentine exposure between individuals of a similar age is not entirely explained by age estimation uncertainty for all four molars”. I think that the use of lowess (and not loess) smoothing regression is not correct in this analysis and do not add any information about the correlation between age at death and dentine exposure. Moreover lowess curves should be plotted with the confidence interval of the locally weighted polynomial regression fit. I suggest the authors to use linear (maybe robust) regression models. � In Figure 4 (now figure 3) the graphs do indeed include a LOESS (not LOWESS, LOWESS means locally weighted and LOESS means locally estimated) regression curve. This display was inspired by a polynomic regression curve on a similar graph in a previous paper by Mays and Pett 2014 cited in our study, and allowed for a visual comparison with medieval samples. You are correct that it does not provide information on correlation, as we specified it was just to illustrate the general trend. However, we have decided to remove these graphs and replace them with linear regressions as suggested as this will be clearer and more concise for many readers. Futhermore we have condensed figure 3 and 4 into a single figure as we believe it conveys the same information, and the infans I and II comparisons are adequately served by table 1.

5. “Based on existing literature, the people buried at Franzhausen I likely began weaning their children before the age of two but a little later than in industrialized populations (75, 76)” this sentence is misleading and wrongly suggests that Humphrey and/or Sellen eventually discussed data about Franzhausen. The onset of weaning is physiologically necessary before one year of age and this well known and discussed by a large literature. � Indeed, we have corrected this sentence and made it clearer. This sentence should have said “weaned” or “finishing weaning” and not “began weaning”. 

6. “Lipid analysis carried out on small ceramic vessels used for infant feeding from the Late Bronze Age and Iron Age in the region confirmed that they contained animal milk (77)” The presence of milk in ceramic vessel for feeding infants DO NOT imply they were weaned. The authors are here confusing the onset of weaning with the use of non human milk in nursing infants. The onset of weaning foresees the introduction of non milk (possibly soft and processed) food into the infant diet. � We have made the reasoning in this section clearer to follow. We did not mention the ceramic vessels to imply that infants were weaned with animal milk, which would indeed be non-sensical, we did so to indicate that there is archaeological evidence that the people of Franzhausen I, contrary to palaeolithic populations for instance, could likely provide animal milk as an alternative to maternal milk and perhaps to help with the weaning process by lessening the burden of breastfeeding on mothers especially as children get older, we believe this information is relevant to a discussion of possible childhood diet in this context.

7. “the observation that the upper and lower dm1 wear appears to start wearing shortly after eruption confirms that weaning had indeed begun by at least 2 years old.” This sentence offers no clues on the nursing in Franzhausen as the onset of weaning before 2 years old is a practice not only well known for all humans (including neanderthals). � We agree that this is an expected result, we have restructured this section to draw more attention to the subject of interest, visible deciduous molar dental wear in young children occurring a little faster than later medieval samples for instance. Of course, it is a shame we cannot say more about weaning, but it is not a central aspect of this study, we only mention it as a starting point for a discussion of childhood diet, we just say what little we can based on our observations to help paint the picture so to speak.

8. Variations in dental wear between male and female children are minor, but there is a slightly higher rate of dentine exposure among younger female children, especially on the lower molars. More wear accumulating on the teeth that erupt first could be explained by differences in diet during early childhood between male and female children, with female children consuming a more abrasive diet (including more abrasives likely found in ground cereals and/or requiring more chewing); alternatively, the weaning process might have started earlier for female children with solid foods being introduced earlier on.” The difference between sexes for dentine exposure il limited to the second lower molar (Ldm2 p = 0.008). But Ldm2s are in occlusion after age 2. Therefore, this unique and not extraordinary evidence tells nothing about differential onset of weaning. � Our reasoning here was that there are multiple explanations for a faster accumulation of dental wear in Ldm2 among young children, different foods are provided to young children based on gender with different impacts on dental wear, or solid foods are introduced earlier into the diet of girls, who would therefore be further along the process with a diet including more frequent solid elements that would produce more wear when the Ldm2 erupts. However, this is just some interesting suggestions/possible explanations for the observed difference and definitely require further investigation using methods more suited to the study of weaning practices to substantiate. So we have altered this section and tried to make our general line of reasoning clearer to follow.

Response to reviewer #1: 

The authors should check carefully the references: there are some missing, some duplications and some in the wrong format.

A major concern is about the statement in Materials and methods: “Many of these deaths were likely due to accidents or rapid infections and not chronic disorders, so the sample is generally representative of children within the community”. It seems to me too speculative, and I don’t see how the authors can declare this. I suggest to the authors to carefully explain and justify this statement otherwise delete it. In any case, I would add a reference to the Osteological paradox. � This statement was meant to convey that we have here a non-catastrophic mortality pattern so the individuals likely did not die as a result of some major crisis, but of course they remain the individuals that died during childhood and as such this statement was incorrect, so we have added a quick discussion of the mortality pattern instead.

The authors should better explain the two surface texture parameters, i.e., complexity and anisotropy, for non-specialists readers and to better understand the discussions. A description of what they are, and the meaning of their variation is in my view necessary. � We have provided more information for understanding the analysis of microwear texture.

Results, Microwear texture analysis: It is hard to me to follow the consideration about the children age and anisotropy and complexity. It is not clear if the age ranges reported are wrong or follow a line of reasoning that I cannot see. Moreover, some of the “low” and “high” reported seems to not correspond to the graph. I suggest to the authors to better explain it and make the data reporting clearer also graphically, this latter to better distinguish between the different individuals. � We have provided more clarity on the microwear texture analysis results to make our reasoning easier to follow.

Discussions: The authors use modern, roman, and medieval samples from previous studies to discuss the Early Bronze Age children from the present study. I agree with this because the lack of comparative data from deciduous teeth from prehistoric samples. Said this, the authors should discuss the chronology of their sample first, highlighting that the differences could be related to the different subsistence economy and to discuss in the dietary interpretation the adult prehistoric diet inferred from dental microwear (Mahoney 2016 reports that no difference are scored about microwear texture analysis between permanent and deciduous teeth). � Concerning the chronology, what information we have is present in the materials section, the site was used over a period of several centuries during which no clear variations in economy or subsistence are known. We will however make clearer that changes may have occurred within the site earlier on. Direct comparisons with adults cannot however be considered reliable, Mahoney’s test although interesting was quite limited in scope (dragging a tooth across an abrasive surface for a fixed distance), we cited this paper stating that the formation of microwear on deciduous and permanent teeth is similar enough to include both dm2 and M1s from children. However, even if the enamel wears in the same way at the microscopic scale, differences in the amount of macroscopic wear, bite force and bite alignment during the mixed dentition phase almost guarantees in our opinion that direct comparisons between adults and children from the same context are unreliable. Comparisons with a broad range of children from other contexts remains therefore in our opinion the best approach for now, even though few studies exist. The idea would be to reveal with each new sample the “landscape” of human, and in this case childhood dental wear in the past.

Discussion, dietary interpretation: Based on existing literature, the people buried at Franzhausen I likely began weaning their children before the age of two but a little later than in industrialized populations. How can the authors support this statement with the data? The method here used is too coarse in time resolution to detect this subtle variation. � We responded to a similar point raised by the editor above. Indeed, the data does not provide this information. This section has been amended to clarify what was meant and where this starting point for our reflection originates from.

Response to reviewer #2: 

There are some issues with the data as it's currently presented that do need to be resolved. In particular the authors need to present the dental pathology data as true prevalence rates - as they currently seem to just be crude prevalence rates, which is inadequate. There also needs to be greater clarity as to what teeth are being recorded. For example, in a child of 8 years with mixed dentition (permanent and deciduous) are only the deciduous teeth included here? If not, then the title of the paper should be altered. Overall, there is a lack of specificity with the dental pathology. � The use of crude prevalence rates was motivated by the wish to keep the data simple and comparable with a broad range of studies including of present-day children, especially as the study mostly focuses on dental wear, however thanks to this insightful comment we have added some true prevalence rates for the caries too, as the number of caries are very low this does not change much to the conclusion, but it can maybe help with future comparisons. We did not use true prevalence rates for the calculus and LEH, as we did not have this data on hand, both a minor variable and so it was not so interesting in the context of this study. We have also made so addition to make it much clearer which teeth are being considered. The word deciduous has been removed from the title as indeed it was a hold over from a previous version that simply evaded our scrutiny and was no longer correct.

I was confused about the point of comparing BA skeletons with present-day (which is so different in terms of sugar consumption it seems a pointless comparison) –� this relates to the frame of our study as established in the introduction which we have also developed more in the abstract, and reflects a choice to use comparisons with present-day individuals as a starting point in our investigation, as they present both the broadest level of interest for the scientific community at large including the medical community, and are in our opinion helpful to anchor ones reflections on living conditions in the past. it's also confusing when the term 'industrialized' is used here when the authors mean present-day rather than 1800s onwards. Technically Europe is 'post-industrial' now. � We would argue that industrial production remains a central aspect of life in present-day Europe, however, this is not the subject of this study, in this context we use the distinction industrialized/pre-industrial that is already well established in previous studies discussing dental wear and pathology cited in the introduction, as the mechanisation of food processing and production is understood to have had a huge impact on both. It would have been much more appropriate to focus on comparisons with other pre-industrial societies. There is some attempt at comparison with a medieval population but the authors should consider the work of Heidi Dawson in more detail, and also the work on deciduous dental pathology by Sian Halcrow - while these are from different periods and places they are still relevant for the discussion. � The absence of more comparisons with pre-industrial societies is definitely and oversight, we have therefore expanded the discussion. We are happy to include further comparisons, as we believe that comparisons with a broad range of children from other context are the best way to understand conditions at Franzhausen I, thank you for the suggestions. We cited a paper from Heidi Dawson earlier in the study but we have also included it in the discussion now. Overall, the fact that the children have a sex estimated and comparisons between boys and girls could be made was useful and novel for a study such as this, although differences are pretty minimal. It would be more informative to see differences between the isotopes between boys and girls than wear. � Our focus was on more mechanical aspects of diet, additionally the preservation of these samples were a huge concern in this study, which is why we opted for non-destructive approach to address the more dietary interpretation related aspects of the study the best we can for the time being, however we do invite future researchers to try and do just this in the discussion, it would certainly be interesting.

Comments

- The title states that this study focuses on the deciduous dentition. The abstract does mentions only 'children'. An age range needs be given as 'children' is a socially specific term. Later in the paper the authors refer to 'under 12s', but these children may have both deciduous and permanent teeth. There needs to be greater assurance that the dental pathologies recorded are only of deciduous teeth (which is not clear). For example, at one point in the paper the authors discuss recording wear on the first permanent molar - so clearly at times teeth other than 'deciduous' are being used. The parameters of this study are very confusing. Please clarify. � We have clarified this in the material a method. Essentially the title should have been amended as it is misleading, permanent teeth were also considered for most measurements (except dentine exposure). 

-Introduction - tell us the date of the site when first introduced. � We have added the information also found in the material and methods.

-Burials are divided into 'infans 1' and 'infans ii' but the age range of each of these categories should be included in the materials and methods text, saying how many individuals in each of the categories. Information for this is only provided later. I also wonder about the choice of these categories given that the discussion centres more on weaning, which will be obscured by the 0-6 year clustering. � We have added 

-Table 1 needs redoing - provide TPRs and provide more detail on the teeth most affected. Are only deciduous teeth included in this table? If permanent are as well then this needs to be made explicit in the discussion. If only deciduous then more information is required. It's only in the discussion that the authors state that caries affected ten molars and one canine in a sample of 1606 teeth - this should be in the results. Why is the LEH not included in the table of pathologies? Further explanation is required. In terms of the intra-populations comparison of pathologies in Table 5 - this should also be TPRs. � All table have been fully redone, both to standardize arrangement, and to make the adjustments made in this comment. We also clarified our statement in the discussion.

- With the dental wear comparisons of younger and older children need to be clear that when refer to molars they mean deciduous - the language needs to be more precise. � We have added more information throughout the text to make this easier to follow which teeth are considered.

-In the discussion the authors suggest that "The results suggest that children before the age of 8 or 9 may have consumed fewer hard food components commonly found outside the domestic setting than their older counterparts". Please explain more clearly what is meant by this. � Some information has been added. Food within the domestic setting can be further processed, grinding, cooking in ceramic pots, and increased consumption harder foods in children has been suggested to relate to the children taken up responsibilities outside the domestic setting and perhaps diversifying their food sources as a result (see Mahoney et al. 2016).

---

## [Decision Letter · Decision Letter 1]

28 Nov 2022

PONE-D-22-19889R1Dental wear and oral pathology among sex determined Early Bronze-Age children from Franzhausen I, Lower AustriaPLOS ONE

Dear Dr. Kanz,

Thank you for submitting your manuscript to PLOS ONE. After careful consideration, we feel that it has merit but does not fully meet PLOS ONE’s publication criteria as it currently stands. Therefore, we invite you to submit a revised version of the manuscript that addresses the points raised during the review process.

The revised paper by Bas et al is an improved version of the original submission and almost all the reviewers’ and mine criticisms have been addressed.

There are still few minor problems that should be addressed by the authors about mostly the weaning sections which is still in my opinion rather speculative and not fully convincing. I suggest the authors to tryn again to make it more consistent or to reduce it to a purely speculative point. The sentence “Previous studies of pre-industrial agrarian populations suggest the people buried at Franzhausen likely finished weaning their children before the age of two but a little later than in industrialized populations [72, 73].” Still do not exclude that the authors of ref 72 and 73 worked on the Franzhausen skeletal series. I suggest to change it into “Previous studies of OTHER pre-industrial agrarian populations COMPARATIVELY suggest …..”.

Abstract (line 40 ff) “Dentine exposure very prevalent in all four deciduous molars affecting over >70% of teeth, with other dental wear measurements indicating a relatively high rate of dental wear.” Please rephrase.

I’m sure that after the amendment of the proposed minor issues, this manuscript will be promptly accepted by Plos One.

We look forward to receiving your revised manuscript.

Kind regards,

Luca Bondioli, M.D.

Academic Editor

PLOS ONE

Journal Requirements:

Reviewers' comments:

Reviewer's Responses to Questions

**Comments to the Author**

1. If the authors have adequately addressed your comments raised in a previous round of review and you feel that this manuscript is now acceptable for publication, you may indicate that here to bypass the “Comments to the Author” section, enter your conflict of interest statement in the “Confidential to Editor” section, and submit your "Accept" recommendation.

Reviewer #1: (No Response)

2. Is the manuscript technically sound, and do the data support the conclusions?

Reviewer #1: Partly

3. Has the statistical analysis been performed appropriately and rigorously? 

Reviewer #1: Yes

4. Have the authors made all data underlying the findings in their manuscript fully available?

Reviewer #1: Yes

5. Is the manuscript presented in an intelligible fashion and written in standard English?

Reviewer #1: Yes

6. Review Comments to the Author

Reviewer #1: The manuscript by Bas and co-authors has been improved after the revision process. The authors carefully considered editor’s and reviewers’ comments, replying to all the issues raised and worked on the text accordingly. Only the weaning issue in the discussion paragraph about Dietary interpretation has not been properly managed. I suggest to the authors to rework this part (Lines 536-540 of the track changes doc) or to delete it (see comment later).

Line 231 (track changes doc): The prevalences of caries, and calculus, and linear….

Add a comma after “caries”

From line 413 (track changes doc): Please check the references #56 #57 #58 in the text (58 appears before 57)

Line 441 (track changes doc): the reference (Duckworth & Huntington; Keyes & Rams, 2016) appears in the text. It needs to be numbered and added to the reference list

Lines 536-540 (track changes doc): In my view the authors didn’t clarify the “weaning issue” here.

I do not agree with the sentence: “Childhood diet begins with the introduction of solid foods in the weaning process”. I would delete this sentence because in my view is wrong. Also, it continues to seem that previous studies refers to Franzhausen people and, as I already said, the method here used is too coarse in time resolution to detect this variation in children diet; moreover, the statement “a little later than in industrialized population” is not supported by data from the present study. Finally, the link with the ceramic baby bottles and the use of animal milk in the weaning process at Franzhausen is totally speculative.

7. PLOS authors have the option to publish the peer review history of their article (what does this mean?). If published, this will include your full peer review and any attached files.

Reviewer #1: No

---

## [Author Response · Author response to Decision Letter 1]

5 Jan 2023

We thank the editor and reviewer for their encouraging feedback and suggested minor corrections. All corrections have been made following your recommendations, most notably we have removed the paragraph on weaning in the discussion as suggested by reviewer 1. Two references have also now been correctly inserted into the reference list and numbered (see reference 63 and 64). We thank you once again for helping us improve our manuscript.

---

## [Editor Report · Decision Letter 2]

8 Jan 2023

Dental wear and oral pathology among sex determined Early Bronze-Age children from Franzhausen I, Lower Austria

PONE-D-22-19889R2

Dear Dr. Kanz,

We’re pleased to inform you that your manuscript has been judged scientifically suitable for publication and will be formally accepted for publication once it meets all outstanding technical requirements.

Kind regards,

Luca Bondioli, M.D.

Academic Editor

PLOS ONE
---

## [Editor Report · Acceptance letter]

27 Jan 2023

PONE-D-22-19889R2 

Dental wear and oral pathology among sex determined Early Bronze-Age children from Franzhausen I, Lower Austria 

Dear Dr. Kanz:

I'm pleased to inform you that your manuscript has been deemed suitable for publication in PLOS ONE. Congratulations! Your manuscript is now with our production department. 

Kind regards, 

on behalf of

Dr. Luca Bondioli 

Academic Editor

PLOS ONE